# Pharmacokinetics and Novel Metabolite Identification of Tartary Buckwheat Extracts in Beagle Dogs Following Co-Administration with Ethanol

**DOI:** 10.3390/pharmaceutics11100525

**Published:** 2019-10-12

**Authors:** Yuancai Liu, Jun Gan, Wanyu Liu, Xin Zhang, Jian Xu, Yue Wu, Yuejun Yang, Luqin Si, Gao Li, Jiangeng Huang

**Affiliations:** 1Hubei Provincial Key Laboratory for Quality and Safety of Traditional Chinese Medicine Health Food, Jing Brand Research Institute, Jing Brand Co. Ltd., Daye 435100, Hubei, China; lyc@jingpai.com (Y.L.); xujian@jingpai.com (J.X.); proyue1122@gmail.com (Y.W.); yyj@jingpai.com (Y.Y.); 2School of Pharmacy, Tongji Medical College, Huazhong University of Science and Technology, 13 Hangkong Road, Wuhan 430030, Hubei, China; jun-cherry@foxmail.com (J.G.); m201875230@hust.edu.cn (W.L.); siriusstar.zx@gmail.com (X.Z.); luqin_si@hust.edu.cn (L.S.); gao_li@hust.edu.cn (G.L.)

**Keywords:** tartary buckwheat extracts, pharmacokinetics, metabolite identification, beagle dog, mass spectrometry

## Abstract

Alcoholic liver disease (ALD) has become a critical global public health issue worldwide. Tartary buckwheat extracts exhibit potential therapeutic effects against ALD due to its antioxidant and anti-inflammatory activities. However, in vivo pharmacokinetics and metabolite identification of tartary buckwheat extracts have not been clearly elucidated. Accordingly, the current manuscript aimed to investigate pharmacokinetics and to identify novel metabolites in beagle dogs following oral co-administration of tartary buckwheat extracts and ethanol. To support pharmacokinetic study, a simple LC-MS/MS method was developed and validated for simultaneous determination of quercetin and kaempferol in beagle dog plasma. The conjugated forms of both analytes were hydrolyzed by β-glucuronidase and sulfatase followed by liquid-liquid extraction using methyl *tert*-butyl ether. In addition, another effective approach was established using advanced ultrafast liquid chromatography coupled with a Q-Exactive hybrid quadrupole orbitrap high resolution mass spectrometer to identify the metabolites in beagle dog biological samples including urine, feces, and plasma. The pharmacokinetic study demonstrated that the absolute oral bioavailability for quercetin and kaempferol was determined to be 4.6% and 1.6%, respectively. Oral bioavailability of quercetin and kaempferol was limited in dogs probably due to poor absorption, significant first pass effect, and biliary elimination, etc. Using high resolution mass spectrometric analysis, a total of nine novel metabolites were identified for the first time and metabolic pathways included methylation, glucuronidation, and sulfation. In vivo pharmacokinetics and metabolite identification results provided preclinical support of co-administration of tartary buckwheat extracts and ethanol in humans.

## 1. Introduction

Alcoholic liver disease (ALD) is a chronic liver disease caused by long term heavy consumption of ethanol. Due to the high morbidity and mortality, it has become a critical global public health issue worldwide [1]. The common spectrum of ALD includes steatosis, fibrosis, cirrhosis, and in some cases hepatic carcinoma. These pathological changes are often triggered by hepatic cell injury, oxidative stress, and inflammation, as well as imbalance of lipid metabolism [2]. As is known, immediate alcohol abstinence is the best therapeutic treatment for all stages of ALD. However, the medical treatment for ALD has not advanced for a long period. As such, effective therapeutic strategies are increasingly required for ALD patients. Intriguingly, a body of research indicates that dietary natural products consisting of flavonoids, resveratrol, and saponins have protective effects against ALD [3].

Buckwheat (*Fagopyrum tataricum*), belonging to the polygonaceae family, is a gluten-free pseudocereal and has been cultivated as crops for centuries. There are mainly four species of buckwheat in the world, namely, sweet buckwheat (*F. esculentum* Moench), tartary buckwheat (*F. tataricum* (L.) Gaertn.), winged buckwheat (*F. emarginatum* Mtissner) and rice buckwheat (*Fagopyrum* spp.). Among those, tartary buckwheat has received increasing attention due to its unique nutritional composition and inspiring health benefits [4,5]. Phytochemical studies have shown that the main chemical components of tartary buckwheat include flavonoids, phenolic compounds, steroids, triterpenoids, amino acids, volatile compounds, etc. [4,5]. Recently, tartary buckwheat extracts demonstrated diverse therapeutic effects, such as antioxidant [6], anti-inflammatory [7], hypoglycemic, and hepatoprotective activities [6]. Moreover, tartary buckwheat extracts were found to have a promising treatment efficacy for ALD in mice by our group (data unpublished).

The bioavailability of dietary components is a vital mediator of their health benefits. As far as is known, there was no published report concerning the pharmacokinetics and bioavailability of tartary buckwheat extracts in beagle dogs. Quercetin and kaempferol are two major flavonoids in tartary buckwheat [5]. Previous studies had convinced that after oral administration of glycosides such as quercitrin, the glycoside forms were subjected to being extensively metabolized during the process of absorption and circulation [8,9]. Therefore, it is imperative to elucidate both the pharmacokinetics and metabolite profiling of glycosides in vivo. It should be noted that there are a large number of aged adults who drink alcohol and take medications simultaneously. Ethanol–drug interaction might alter therapeutic effectiveness or result in severe adverse reactions due to changed absorption, distribution and elimination of alcohol and medications [10]. Moreover, concern has risen over the interactions between alcohol and drugs in patients with ALD [11,12]. Therefore, the concomitant ingestion of alcohol may alter pharmacokinetic processes of tartary buckwheat extracts. Due to its important social and medical implications, ethanol mediated pharmacokinetic alteration is worthy of investigation. As such, the current work aimed to investigate the pharmacokinetics and in vivo metabolites of tartary buckwheat extracts in beagle dogs following co-administration with ethanol using both liquid chromatography-tandem mass spectrometry (LC-MS/MS) and ultrafast liquid chromatography (UFLC)-Q-Exactive hybrid quadrupole orbitrap high resolution mass spectrometry (HRMS) platforms.

## 2. Materials and Methods 

### 2.1. Chemicals and Reagents

The reference standards of quercetin, kaempferol, rutin, kaempferol-3-O-rutinoside, isoquercitrin, hyperoside, *N*-*trans*-Feruloyltyramine, and astragalin were purchased from the National Institute for the Control of Pharmaceutical and Biological Products (Beijing, China). Internal standard (IS) d3-quercetin was obtained from Toronto Research Chemicals (North York, ON, Canada). Sulfatase (type H-1) and β-glucuronidase (type H-2) were obtained from Sigma-Aldrich (St Louis, MO, USA). Deionized water was freshly prepared using a Milli-Q purification system (Millipore, Billerica, MA, USA). HPLC-grade acetonitrile and methanol were purchased from Fisher Scientific (Fair Lawn, NJ, USA). EDTA-K_2_ containing vacutainer tubes (5 mL) and 22 G IV catheters (0.9 × 25 mm) were obtained from BD Biosciences (Franklin Lakes, NJ, USA). Formic acid, methyl *tert*-butyl ether (MTBE), and other chemical reagents were of analytical grade. The tartary buckwheat extracts were produced by Jing Brand Co. Ltd. (Daye, China). The contents of rutin, quercetin, kaempferol and, kaempferol-3-O-rutinoside were 61.2%, 4.5%, 0.15%, and 4.84% respectively, which was quantified by HPLC.

### 2.2. Instrumentation and Analytical Conditions

#### 2.2.1. LC-MS/MS Quantitative Analysis Conditions

A high-performance liquid chromatographic system consisting of two LC-20AD binary pumps, a DGU-20A5 degasser, a SIL-20AC auto-sampler, and a CTO-20A column oven (Shimadzu, Kyoto, Japan) was used throughout. Chromatographic separation was performed on a Welch Ultimate^®^ XB C18 column (2.1 × 50 mm, 5 μm, Welch Materials Inc., Shanghai, China) under a column temperature of 40 °C. The mobile phase consisted of 0.1% formic acid in water (A) and acetonitrile (B) using the following gradient condition: 10% B at 0–1.0 min, 60% B at 1.01–3.3 min, and 10% B at 3.31–5.0 min. The flow rate was set at 0.4 mL/min and the sample injection volume was 10 μL.

Mass spectrometric detection was carried out on an API 4000 QTrap^®^ triple quadrupole mass spectrometer (Applied Biosystem/MDS Sciex, Foster City, CA, USA) equipped with an electrospray ionization (ESI) source. Multiple reaction monitoring was used for MS/MS analyses under negative ionization mode and the optimal MS parameters were used as follows: ion source temperature, 500 °C; ion spray voltage, −4500 V; curtain gas, 25 psi; collision gas, medium; gas 1, 40 psi; and gas 2, 50 psi. The quantitative ion pairs of analytes and IS and their optimal compound parameters including declustering potential, collision energy, and cell exit potential are listed in Appendix A. Analyst software (Version 1.6.1, Applied Biosystem/MDS Sciex, Foster City, CA, USA) was used for analytical data acquisition and processing.

#### 2.2.2. UFLC-Q-Exactive Orbitrap HRMS Qualitative Analysis Conditions

A Shimadzu Prominence UFLC system consisting of two LC-30AD binary pumps, a SIL-30AC auto-sampler, a DGU-20A5 degasser, and a CTO-30A column oven (Shimadzu Corporation, Kyoto, Japan) was used throughout. Chromatographic separation was carried out on a BEH C18 column (2.1 × 100 mm, 1.7 µm, Waters Co., Milford, MA, USA). The mobile phase consisted of water containing 0.1% (*v/v*) acetic acid (A) and acetonitrile containing 0.1% (*v/v*) acetic acid (B). The gradient program was: 0–20 min at 15–55% B; 20–45 min at 55–95% B; 45–53 min at 95% B; 53–54 min at 95–15% B; and 54–60 min at 15% B. The flow rate was set at 0.3 mL/min and the LC column temperature was maintained at 40 °C. The sample injection volume was 10 µL.

Mass spectrometric analysis was performed with a Q-Exactive hybrid quadrupole orbitrap high resolution mass spectrometer (Thermo Fisher Scientific, Bremen, Germany) using a heated electrospray ionization source in the positive ionization mode. The operating parameters were as follows: spray voltage, 3.5 kV; sheath gas pressure, 35 arbitrary units (arb); auxiliary gas pressure, 10 (arb); capillary temperature, 320 °C; auxiliary gas heater temperature, 350 °C; scan modes, full MS (resolution 60,000) and scan range, *m/z* 200–1200. All data were acquired using the Xcalibur 2.0 software (Thermo Fisher Scientific Inc., Waltham, MA, USA). 

### 2.3. Preparation of Stock Solutions, Calibration Standards, and Quality Control (QC) Samples

The stock solutions of quercetin, kaempferol, and IS were individually prepared by dissolving a certain amount of reference standards in methanol to give a final concentration of 1 mg/mL. The mixed quercetin and kaempferol working solutions were obtained by serial dilution of stock solutions using methanol-water (1:1, *v/v*) to obtain concentrations of 0.01, 0.02, 0.1, 1, 4, 10, 18, and 20 μg/mL, while the working solution of IS was diluted to 2 μg/mL. All the standard solutions were stored at −80 °C until use.

To prepare calibration standards, 5 μL working solutions were spiked with 95 μL aliquots of pooled blank plasma obtained from six individual beagle dogs, receiving a series of calibration standards with final concentrations of 0.5, 1, 5, 50, 200, 500, 900, and 1000 ng/mL for each analyte. QC samples were obtained in a similar manner at three different levels of 1.5, 30, and 750 ng/mL, corresponding to low quality control (LQC), medium quality control (MQC), and high quality control (HQC), respectively. Then, 100 μL aliquots of calibration standards and QC samples were spiked with 10.5 μL of 0.5 M acetic acid containing 2 mg/mL ascorbic acid and stored at −80 °C until use.

### 2.4. Sample Preparation

#### 2.4.1. Sample Preparation for Quantitative Analysis

After thawing on wet ice, 5 μL IS working solution was added to 110.5 μL aliquots of calibration standards and QC samples spiked with acetic acid and ascorbic acid. Actual plasma samples for quantitative determination were required to be incubated with 200 units of each of sulfatase and β-glucuronidase for 30 min prior to the addition of IS. Thereafter, 1000 μL MTBE was added to extract both analytes. The samples were vortexed for 10 min and centrifuged at 15,000 rpm for 10 min at 4 °C. Then, 800 μL supernatant was transferred and evaporated to dryness under vacuum using an SPD1010 SpeedVac Concentrator (Thermo Fisher Scientific, Waltham, MA, USA) at 45 °C, 5.1 Vac for 2 h. The residue was reconstituted with 80 μL mobile phase. After centrifugation at 13,000 g for 10 min at 4 °C, 10 μL supernatant was injected for LC-MS/MS analyses.

#### 2.4.2. Sample Preparation for Qualitative Analysis

A dried faecal sample was extracted with acetonitrile (1:6, *w/v*) by ultrasonication for 20 min. The extraction was centrifuged at 1500 g for 10 min at 4 °C, the supernatant was evaporated using an SPD1010 SpeedVac Concentrator at 45 °C, 5.1 Vac for 2 h. The residue was reconstituted in 100 μL of 50% methanol in water, and then centrifuged at 13,000 g for 10 min at 4 °C. The supernatant was transferred into a vial for analysis. 

Urine and plasma samples were extracted with acetonitrile (1:6, *v/v*). After vortex for 3 min and centrifugation at 1500 g for 10 min at 4 °C, the supernatant was evaporated to dryness using an SPD1010 SpeedVac Concentrator at 45 °C, 5.1 Vac for 2 h. The residue was reconstituted in 100 μL of 50% methanol in water, and then centrifuged at 13,000 g for 10 min at 4 °C. The supernatant was transferred into a vial for analysis.

### 2.5. Method Validation

The method was fully validated according to guidelines for quantitative analysis of biological samples. The selectivity for quercetin and kaempferol was evaluated to exclude interference from endogenous substances. The linearity of standard curves was validated by plotting the peak area ratios of each analyte to IS versus the nominal concentrations ranging from 0.5 ng/mL to 1000 ng/mL for quercetin and kaempferol. The lower limit of quantification (LLOQ) was defined as the lowest concentration of the calibration curve with accuracy within ±20% and precision <20%. The accuracy and precision of the methods were assessed by measuring the LLOQ, LQC, MQC, and HQC samples in six replicates with three consecutive analytical runs. The matrix effects and extraction recoveries of quercetin and kaempferol were evaluated by determining six replicates of QC samples at LQC, MQC, and HQC levels. The dilution reliability was evaluated by diluting the samples spiked at 4000 ng/mL quercetin and kaempferol with blank plasma with a ratio of 1:5 to reach a final concentration of 800 ng/mL. After ULOQ sample analysis, blank samples were injected to estimate carry-over. The stability of quercetin and kaempferol in biological matrices was evaluated by analyzing six replicates of QC samples at LQC, MQC, and HQC levels under various conditions including four freeze-thaw cycles, 24 h storage at room temperature and 30-day storage at −80 °C, respectively. The post-preparative stability was determined at auto-sampler 4 °C for 24 h.

### 2.6. Pharmacokinetics and Metabolite Identification in Beagle Dogs

Six healthy beagle dogs (10 ± 1 kg), male and female in half, were purchased from Hubei Yizhicheng Biotechnology Co. Ltd. (Wuhan, China). The dog was housed in a stainless-steel canine cage with free access to water. The animal experiments were conducted in accordance with the Guidelines for Care and Use of Laboratory Animals and approved in June 2017 by the Animal Care and Ethics Committee of Tongji Medical College at Huazhong University of Science and Technology (Project ID.: HUST-D-201706112).

For the pharmacokinetic study, the dogs were given a single oral dose of 100 mg/kg tartary buckwheat extracts suspended in 50% (*v/v*) ethanol. After a 10-day washout period, the dogs were intravenously administered at a dose of 5 mg/kg tartary buckwheat extracts dissolved in 50% ethanol and filtered through a 0.22 μm sterile filter membrane. Blood samples (approximately 2 mL) were collected from the forelimb vein into 5 mL EDTA-K_2_ vacuum collective tubes at predose and 0.08, 0.17, 0.33, 0.5, 0.75, 1, 1.5, 2, 2.5, 3, 3.5, 4, 6, 8, 10, and 12 h, respectively. Blood samples were stored on wet ice and centrifuged at 3500 rpm for 10 min on 4 °C to obtain plasma. 400 μL of each plasma samples were spiked with 42 μL of 0.5 M acetic acid containing 2 mg/mL ascorbic acid and divided into two equal parts for quantitative and qualitative analyses, respectively. Blood samples collected at predose and 0.5, 1, 2, and 4 h were used for metabolite identification. The urine and feces samples were collected every 12 h for up to 48 h. The biological samples were all stored at −80 °C until assay. The pharmacokinetic parameters of quercetin and kaempferol were calculated by Drug and Statistics (DAS) software (Version 2.1.1, Mathematical Pharmacology Professional Committee of China, Shanghai, China). Noncompartmental analysis was used to calculate pharmacokinetic parameters. The maximum plasma concentration (C_max_) and time to reach C_max_ (T_max_) were obtained directly from the plasma concentration versus time curve, the area under the plasma concentration versus time curve (AUC) was calculated using the linear trapezoidal method, and the elimination half-life (t_1/2_) was estimated by 0.693/k. For intravenous administration, the apparent volume of distribution (Vd) was determined using equation Vd = (Dose/AUC)/k and apparent clearance (CL) was calculated as the ratio of dose to AUC. The dose was corrected by the content of quercetin and kaempferol derived components. The absolute bioavailability (F) was calculated as the ratio of oral AUC to the intravenous AUC.

## 3. Results and Discussion

### 3.1. Method Development

In most cases, flavonoids such as quercetin and resveratrol could be converted into glucuronidated and sulfated conjugates in vivo. As a result, the total concentration of both unconjugated and conjugated flavonoids was generally determined after hydrolysis. Various hydrolysis methods using β-glucuronidase and/or sulfatase was employed to transform glucuronidated and sulfated conjugates to the corresponding free forms [13]. In previous methods, the amount of β-glucuronidase and sulfatase varied among different procedures of enzymatic hydrolysis, likely due to diverse sources of biological samples with varying concentrations of glucuronide and sulfate conjugates as well as different enzymatic activities of β-glucuronidase and sulfatase. For example, Lee et al. employed an enzymatic hydrolysis method for quercetin glucuronide and sulfate conjugates using 150 units of sulfatase and 17,000 units of glucuronidase [14], while another hydrolysis procedure using 2.5 units of sulfatase and 10 units of glucuronidase was reported [15]. To the best of our knowledge, this was the first report to evaluate hydrolysis effectiveness of quercetin and kaempferol conjugates in beagle dog plasma under varying amount of β-glucuronidase, sulfatase, or the combination of β-glucuronidase and sulfatase. As shown in Figure 1, both β-glucuronidase and sulfatase demonstrated an increasing hydrolysis potency for quercetin conjugates with individual enzyme amount (from 1 unit to 200 units), but tended to reach a plateau at 200 units. Similar results were observed for kaempferol. As expected, mixed hydrolase exhibited higher hydrolysis potency than the single enzyme alone. Finally, mixed hydrolase, including 200 units of each enzymes, was selected to ensure that the conjugated metabolites of quercetin and kaempferol can be completely hydrolyzed in beagle dog plasma.

Stabilizers are often required to improve the stability of flavonoids of interest [16]. According to previous reports, the pH of biological samples was maintained at pH5 using acetic acid [15]. Then, the stability of quercetin and kaempferol was assessed after adding different stabilizers such as ascorbic acid, sodium bisulfite, or combination of ascorbic acid and citric acid followed by 2 h incubation under room temperature. The results are shown in Table 1. Quercetin and kaempferol were not stable by only controlling the pH, but the remaining three strategies were able to stabilize both analytes of interest. Finally, 0.5 M acetic acid containing 2 mg/mL ascorbic acid was chosen as stabilizer. Thereafter, the liquid-liquid extraction (LLE) method employing five different organic extractants including ethyl acetate, MTBE, dichloromethane, n-hexane, and ether was tested. The results are shown in Figure 2. Based on a matrix effect and extraction recovery, MTBE was selected as the extraction solvent for sample preparation.

### 3.2. Method Validation

The selectivity of quercetin and kaempferol was evaluated using six individual beagle dogs’ plasma and the typical MRM chromatograms are shown in Appendix A. Each calibration curve was analyzed using the least square linear regression equation with a weighting factor of 1/x^2^. The linear ranges and correlation coefficients of quercetin and kaempferol are listed in Appendix A, respectively. The calibration curves of both analytes exhibited good linear response over the concentration range of 0.5–1000 ng/mL. In addition, the LLOQ reached down to 0.5 ng/mL for both quercetin and kaempferol. The results of intraday and interday accuracy (RE) and precision (RSD) were within ±15% for both analytes of interest (Appendix A). The intraday and interday accuracy was with ±15% for both analytes of interest. The matrix effect and extraction recovery of quercetin, kaempferol, and IS are shown in Appendix A. The results indicated that there was no significant matrix effect under the current sample processing, chromatographic, and mass spectrometric conditions. The extraction recoveries were found to be similar among both analytes and IS. The results of dilution reliability showed that the accuracy and precision of quercetin and kaempferol during dilution process both met the acceptance criteria (Appendix A). Also, there was no obvious carry-over effect (data not shown). The stability results indicated that quercetin and kaempferol were stable in beagle dog plasma under different conditions such as four freeze-thaw cycles at −80 °C, short term room temperature storage for 24 h, long term stability under −80 °C for 30 days, and postpreparative storage at 4 °C for 24 h (Appendix A). Also, each stock and working solution was stable after storage at −80 °C for 30 days when compared with freshly prepared stock and working solutions (data not shown).

### 3.3. Pharmacokinetic Study

In the current work, a reliable LC-MS/MS method was established and validated for the pharmacokinetic and bioavailability study of quercetin and kaempferol in beagle dogs. The plasma concentration versus time curves of total quercetin and kaempferol in beagle dogs are presented in Figure 3. The major pharmacokinetic parameters are listed in Table 2. Quercetin exhibited higher systemic exposure than kaempferol, which agreed with the composition of quercetin and kaempferol in tartary buckwheat extracts. As expected, quercetin and kaempferol demonstrated alike volumes of distribution after intravenous administration due to the similar structures between both compounds of interest. Interestingly, kaempferol showed significantly larger clearance than quercetin probably owing to distinct enzymatic metabolism and transporter-mediated excretion. The absolute oral bioavailability was found to be 4.6% and 1.6% for quercetin and kaempferol, respectively. In comparison with pharmacokinetic parameters of quercetin and kaempferol administered with control vehicle, co-administration with ethanol increased the C_max_ and shortened T_max_ after oral administration. As a result, the bioavailability was increased by around 40%, which might be explained by altered solubility, intestinal absorption, and metabolic activity of both intestinal and hepatic enzymes. Our findings were consistent with previously reported bioavailability of quercetin in beagle dogs [17,18]. However, compared to human study, in which the absolute bioavailability of quercetin was surprisingly estimated as 44.8% based on total radioactivity, the absolute bioavailability was likely to be underestimated in beagle dogs since methylated and oxidized forms were not considered [19]. It has been reported that co-administration with ethanol raised the intestinal absorption of quercetin in a concentration dependent manner [20] and increased systemic exposure of kaempferol along with an obvious decrease in total body clearance in rats [21]. In addition, the ethanol metabolite acetaldehyde was reported to alter the oral bioavailability of therapeutic molecules through modulating tight junctions and paracellular permeability [22]. However, oral bioavailability of quercetin and kaempferol was limited in dogs due to poor absorption, significant first pass effect, and biliary elimination, etc. [23]. In summary, co-administration with ethanol might alter the pharmacokinetics and bioavailability of quercetin and kaempferol to a certain extent.

### 3.4. Metabolite Identification

In previous study, 16 metabolites have been detected in rat biological matrices including feces, urine, bile, and plasma after oral administration of 240 mg/kg tartary buckwheat extracts [24]. However, the metabolite profiling of tartary buckwheat extracts in beagle dogs has not been elucidated until now. In the current study, a dose of 100 mg/kg tartary buckwheat extracts was co-administered with ethanol to beagle dogs. Metabolites were identified according to the following steps: (1) chemical database construction including mass weights, elemental compositions, and structure information of chemical components originated from tartary buckwheat extract; (2) understanding of the mass fragmentation information of the typical chemical components in tartary buckwheat extract; (3) screening of potential metabolites by comparing the mass chromatograms and characteristic fragment ions between the dosing and blank control group. In addition to reported metabolites such as methylated, glucuronidated and sulfated quercetin and kaempferol [18,25], a total of nine novel metabolites were detected and further characterized using the above-mentioned strategy. Three novel metabolites were identified and characterized in each biological matrix, including urine (M1, M7, and M8), feces (M3, M5 and M9), and plasma (M2, M4, and M6). The measured masses, mass errors, characteristic product ions of the proposed metabolites, and predicted metabolic pathways are summarized in Table 3. The MS/MS (MS^2^) spectra of metabolites detected in urine, feces, and plasma are shown in Figure 4.

Metabolite M1 (t_R_ = 3.23 min) exhibited a protonated molecule [M + H]^+^ ion at *m/z* 611.1613 (calculated for C_26_H_26_O_17_). The MS^2^ spectra yielded an ion at *m/z* 479 due to the loss of xyloside (C_5_H_8_O_4_, 132 Da). The typical fragment ion at *m/z* 303 was produced with the highest relative intensity by successive loss of glucuronide (C_6_H_8_O_6_ 176 Da). In addition, the product ions at *m/z* 285 [303–H_2_O], *m/z* 239 [303–2H_2_O–CO], and *m/z* 221 [303–3H_2_O–CO], which were similar to the fragments of quercetin based on the reference standard (Appendix A). These fragment ions provided substantial evidence for indicating that M1 was tentatively identified as quercetin xyloside glucuronide.

M2 (t_R_ = 8.70 min) and M6 (t_R_ = 12.10 min), which produced [M + H]^+^ ion at *m/z* 677.3398 (calculated for C_26_H_28_O_19_S), were preliminarily thought as sulfated products due to a neutral loss of 80 Da. In the MS^2^ spectrum, the [M + H]^+^ ion generated typical fragment ions at *m/z* 465 [677–xyloside–sulfate], *m/z* 383 [667–xyloside–glucoside], *m/z* 339 [677–xyloside–sulfate–C_6_H_6_O_3_], and *m/z* 126 (the highest relative intensity) [C_6_H_6_O_3_]. Additionally, a battery of fragments at *m/z* 302, 257 [303–CO–H_2_O], 243 [303–CO–H_2_O–CH_2_], 229 [303–2CO–H_2_O], 213 [303–2CO–H_2_O–O], 201 [303–3CO–H_2_O], and 139 [C_7_H_7_O_3_] were identified to be characteristic fragment ions of quercetin. Therefore, M2 and M6 were identified as quercetin xyloside glucoside sulfate.

M3 (t_R_ = 8.72 min) showed [M + H]^+^ ion at *m/z* 493.1615 (calculated for C_23_H_24_O_12_). In the MS^2^ spectrum, the [M + H]^+^ ion produced typical fragment ions at *m/z* 465 [493–2CH_2_], *m/z* 303 [493–2CH_2_–glucoside]. The characteristic fragment ions at *m/z* 285 [303–H_2_O], 257(the highest relative intensity) [303–CO–H_2_O], 229 [303–2CO–H_2_O], 219 [303–3CO], and 165 [303–CO–C_6_H_6_O_2_] were explored, which were in accordance with the mass fragmentation behavior of quercetin. Besides, the presence of fragment ions at *m/z* 165 indicated that dimethylation had occurred at C-3′ or C-4′ position of B-ring (Appendix A). M3 was identified as dimethyl quercetin glucoside.

M4 (t_R_ = 10.06 min) showed [M + H]^+^ ion at *m/z* 719.3624 (calculated for C_29_H_34_O_19_S). In the MS^2^ spectrum, the fragments consisted of *m/z* 611 [719–2CH_2_–sulfate], *m/z* 465 [719–2CH_2_–sulfate–rhamnoside], *m/z* 339 [719–2CH_2_–sulfate–C_6_H_6_O_3_], and *m/z* 126 (the highest relative intensity) [C_6_H_6_O_3_]. The other fragments at *m/z* 257, 243, 229, 213, 201, 135, and 109 were consistent with product ions of quercetin. Thus, M4 was identified as dimethyl quercetin rhamnoside glucoside sulfate.

M5 (t_R_ = 11.25 min) gave rise to protonated ion at *m/z* 557.4034 (calculated for C_23_H_24_O_14_S). In the MS^2^ spectrum, the fragments were *m/z* 383 [557–2CH_2_–rhamnoside], *m/z* 331 [557–rhamnoside–sulfate], and *m/z* 303 [557–2CH_2_–rhamnoside–sulfate]. Moreover, the fragment ions at *m/z* 285, 275, 257, 229, 219, 193 (the highest relative intensity), 177, 165, and 153 were characteristic ions of quercetin. Among those, the fragment ions at *m/z* 165 and 153 were from the A-ring of quercetin aglycone, while the fragments *m/z* 193 and 177 were found 28 Da and 14 Da more than 165, respectively, indicating that the methylation might occur at A or C ring (Appendix A). Therefore, M5 was identified as dimethyl quercetin rhamnoside sulfate.

M7 (t_R_ = 12.83 min) displayed [M + H]^+^ ion at *m/z* 831.5027 (calculated for C_34_H_38_O_24_). In the MS^2^ spectrum, the fragments included *m/z* 637 [831–H_2_O–glucuronide], *m/z* 461 [831–H_2_O–2glucuronide], and *m/z* 317 [831–glucoside (162 Da)–2 glucuronides (176 Da)]. The typical fragment ions at *m/z* 389, 371, 353 (the highest relative intensity), 335, and 317 were due to the gradual loss of H_2_O of quercetin glycoside. Besides, the MS^2^ spectrum illustrated that the fragment ions at *m/z* 303, 285, 271, 243, 227, 209, and 137 were characteristic fragment patterns of quercetin. Therefore, M7 was identified as methyl quercetin glucoside diglucuronide.

M8 (t_R_ = 13.76 min) produced protonated [M + H]^+^ ion at *m/z* 331.2061 (calculated for C_17_H_14_O_7_), which was 28 Da higher than [M + H]^+^ ion for quercetin. The major fragment ion at *m/z* 135 with the highest relative intensity was attributed to the loss of CO and H_2_O from *m/z* 181 (C_8_H_5_O_5_) which was firstly generated due to the Retro Diels-Alder reaction of quercetin. Precursor ion at *m/z* 331 also produced a battery of characteristic fragment ions at *m/z* 317, 313, 303, 275, 229, 181, 135, and 121. In addition, the presence of fragment ions at *m/z* 181 and 135 indicated that the biotransformation might occur at C-3′ or C-4′ position of the B ring. M8 was tentatively identified as dimethyl quercetin.

M9 (t_R_ = 15.56 min) showed [M + H]^+^ ion at *m/z* 609.3254 (calculated for C_28_H_32_O_15_). Interestingly, the protonated ion *m/z* 609 produced a series of fragment ions at *m/z* 591 [609–H_2_O], 577 [591–CH_2_], 549 [577–CO], 531[549–H_2_O], 485 [531–CO–H_2_O], 467 [485–H_2_O], and 449 [467–H_2_O], indicating that the fragmentation pattern was dominated by the internal fracture of the sugar ring instead of the loss of whole sugar unit. In addition, the fragment ions at *m/z* 549 [609–C_2_H_4_O_2_ (60 Da)] and 359 [449 (quercetin glycoside)–C_3_H_6_O_3_ (90 Da)] were also found. The loss of the typical fragments 60 and 90 Da suggested that the substituents on the parent nucleus may be pentose rather than hexose [26]. Taken together, M9 was proposed as a flavone glycoside substituted by pentacarbose. Moreover, the fragment ions at *m/z* 317, 331, and 345 were 14, 28, and 42 Da higher than *m/z* 303 (the highest relative intensity), respectively. The characteristic ions at *m/z* 285, 275, 257, 229, and 165 were in good agreement with those of quercetin. Thus, M9 was identified as trimethyl quercetin di-C-xyloside.

## 4. Conclusions

This study established a reliable LC-MS/MS method for simultaneous determination of quercetin and kaempferol in dog plasma. The method was fully validated and successfully applied for pharmacokinetic and bioavailablity study of tartary buckwheat extracts in beagle dogs following co-administration with ethanol. Also, another effective UFLC-Q-Exactive hybrid quadrupole orbitrap HRMS approach was established to identify the metabolites in beagle dog urine, feces, and plasma. The pharmacokinetic results showed that the bioavailability of quercetin and kaempferol was less than 5% probably due to poor absorption, significant first pass effect, and biliary elimination, etc. A total of nine novel metabolites were identified for the first time and metabolic pathways included methylation, glucuronidation, and sulfation. In vivo pharmacokinetics and metabolite identification results provided preclinical support of co-administration of tartary buckwheat extracts and ethanol in humans.

## Figures and Tables

**Figure 1 pharmaceutics-11-00525-f001:**
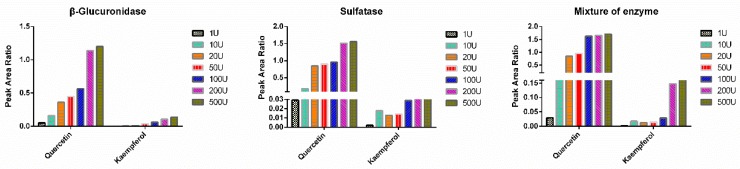
Peak area ratio of analytes with different volume of hydrolase.

**Figure 2 pharmaceutics-11-00525-f002:**
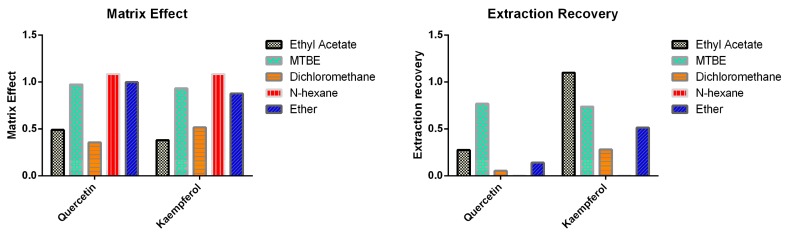
Comparison of matrix effect and extraction recovery of different extractants based liquid-liquid extraction method. Methyl *tert*-butyl ether, MBTE.

**Figure 3 pharmaceutics-11-00525-f003:**
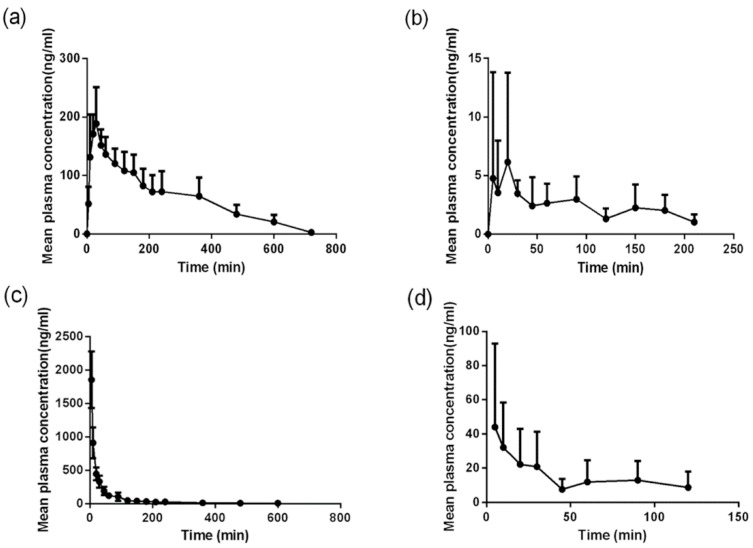
The mean total plasma concentration versus time curves of (**a**,**c**) quercetin and (**b**,**d**) kaempferol in beagle dogs following either (**a**,**b**) oral (100 mg/kg) or (**c**,**d**) intravenous (5 mg/kg) administration of tartary buckwheat extracts. Data are represented in mean ± S.D., *n* = 6.

**Figure 4 pharmaceutics-11-00525-f004:**
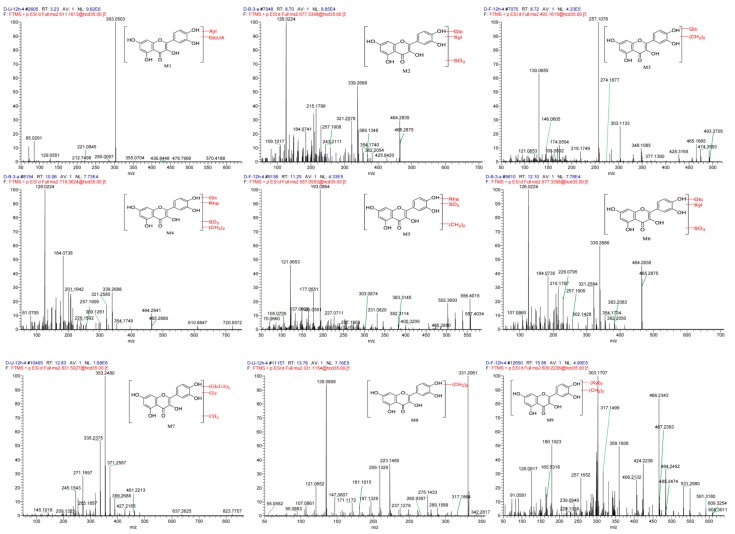
MS/MS spectra of novel metabolites detected in beagle dog urine, feces, and plasma.

**Table 1 pharmaceutics-11-00525-t001:** Plasma stability at room temperature for 2 h of quercetin and kaempferol after using different stabilizers.

Analyte	QC Level	Response Difference (%)			
		No Stabilizer	Ascorbic Acid	Sodium Bisulfite	Ascorbic Acid + Citric Acid
quercetin	LQC (1.5 ng/mL)	−29.80	−5.62	−6.46	−1.03
	HQC (750 ng/mL)	−21.50	1.22	−5.94	1.62
kaempferol	LQC (1.5 ng/mL)	−19.73	−10.50	5.10	1.28
	HQC (750 ng/mL)	−12.45	0.98	−2.70	−1.30

**Table 2 pharmaceutics-11-00525-t002:** Pharmacokinetic parameters of total quercetin and kaempferol in beagle dogs following oral or intravenous administration of tartary buckwheat extract. Data are represented in mean ± S.D., *n* = 6.

Parameter	Unit	Oral Administration (100 mg/kg)		Intravenous Administration (5 mg/kg)	
		Quercetin	Kaempferol	Quercetin	Kaempferol
AUC_0–t_	ng/mL·min	44116 ± 7408	568.2 ± 269.4	48368 ± 8439	1809.6 ± 1702.6
t_1/2_	min	228.6 ± 140.0	102.1 ± 87.0	194.9 ± 81.5	47.9 ± 32.2
T_max_	min	26.7 ± 17.5	38.2 ± 32.3	-	-
C_max_	ng/mL	222.8 ± 53.1	9.5 ± 8.6	-	-
CL	L/min/kg	-	-	0.07 ± 0.01	0.28 ± 0.24
Vd	L/kg	-	-	19.3 ± 8.9	20.6 ± 12.5
F	%	4.6	1.6	-	-

**Table 3 pharmaceutics-11-00525-t003:** Identification of 9 novel metabolites in beagle dog urine, feces, and plasma.

No.	Rt (min)	Formula	[M + H]^+^ (m/z)	Error (ppm)	Fragment Ions (*m/z*)	Source	Identification	Metabolic Pathway
M1	3.23	C_26_H_26_O_17_	611.1613	0.6832	479, 303, 285, 239, 221	U.	Quercetin xyloside glucuronide	Glucuronidation
M2	8.70	C_26_H_28_O_19_S	677.3398	−2.9961	465, 369, 339, 321, 302, 257, 243, 229, 213, 201,139	P.	Quercetin xyloside glucoside sulfate	Sulfation
M3	8.72	C_23_H_24_O_12_	493.1615	−0.5376	465, 303, 285, 257, 229, 219, 165	F.	Dimethyl quercetin glucoside	Dimethylation
M4	10.06	C_29_H_34_O_19_S	719.3624	−4.4449	611, 465, 339, 257, 243, 229, 213, 201, 135, 126, 109	P.	Dimethyl quercetin rhamnoside glucoside sulfate	Dimethylation, Sulfation
M5	11.25	C_23_H_24_O_14_S	557.4034	−0.1276	539, 521, 503, 383, 331, 303, 285, 275, 257, 229, 219, 193, 177, 165, 153	F.	Dimethyl quercetin rhamnoside sulfate	Dimethylation, Sulfation
M6	12.10	C_26_H_28_O_19_S	677.3398	−2.9060	465, 369, 339, 321, 302, 257, 243, 229, 213, 201, 139	P.	Quercetin xyloside glucoside sulfate	Sulfation
M7	12.83	C_34_H_38_O_24_	831.5027	−0.3240	137, 209, 227, 243, 271, 285, 303, 317, 335, 353, 371, 389, 461, 637	U.	Methyl quercetin glucoside diglucuronide	Methylation, Diglucuronidation
M8	13.76	C_17_H_14_O_7_	331.2061	−0.1946	317, 313, 303, 289, 275, 229, 181, 135, 125	U.	Dimethyl quercetin	Dimethylation
M9	15.56	C_28_H_32_O_15_	609.3254	−0.2484	591, 577, 549, 485, 467, 449, 359, 345, 331, 317, 303, 531, 285, 275, 257, 229, 165	F.	Trimethyl quercetin di-C-xyloside	Trimethylation

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
