# Peer review of "Pharmacokinetics and Novel Metabolite Identification of Tartary Buckwheat Extracts in Beagle Dogs Following Co-Administration with Ethanol"

_pharmaceutics, 2019, doi:10.3390/pharmaceutics11100525_

Round 1

Reviewer 1 Report

This manuscript has been well described, but I consider that some modifications are needed and a few comments are included in this review to improve the study report.

In Table 2, additional PK parameter clearance (CL), distribution volume (Vd), and bioavailability (BA and F) should be included and please discuss it. The Cmax after i.v. administration in the Table 2 should be excluded. The values has been shown as mean +/- SD or SE? Please describe it. 

Author Response

According to the reviewer’s comment, these PK parameters including clearance (CL), distribution volume (Vd) and bioavailability (F) have been included in Table 2. Also, detailed discussion concerning these parameters have been added in revised manuscript. The Cmax after intravenous administration in the Table 2 has been excluded. In addition, the values are presented as mean +/- SD. This information has been included in Table and Figure legend.

Reviewer 2 Report

Authors report a pharmacokinetics study of tartary buckwheat extracts in beagle dogs. They report an abosolute oral bioavailability of quercetin and kaempferol to be 4.6% and 1.6% and 9 novel metabolites were identified. Manuscript reports the methods, results and conclusion in appropriate way.

Author Response

Thank you very much for your effort on our manuscript.

Reviewer 3 Report

Dear the Editor

Liu Y et al studied the pharmacokinetics of several components in tartary buckwheat extracts in beagle dog. Eventually these authors uncovered 9 novel metabolites when ethanol was co-administered in this animal model. To study more in detail, the metabolism of quercetin and kaempferol was examined. Interestingly, the bioavailability of them was very low: 4.6% for quercetin and 1.6% for kaempferol. These authors argued that poor absorption, significant first pass effect, and biliary elimination could be the three major reasons for this low bioavailability. The identified reactions of novel metabolites include methylation, glucuronidation, and sulfation as determined by high resolution mass spectrometry. These results suggest that co-administration of ethanol and tartary buckwheat might induce previously unappreciated metabolic reactions under pathophysiological conditions such as alcoholic liver disease.

Comments:

This is a well-written manuscript for pharmacokinetic study in animal model. The method validation has been done properly. Although previous study has been performed in rat, these authors proposed that the metabolism of tartary buckwheat in beagle dog might be distinct from rodents.

Minor concerns:

1) In page 4, line 156, “vortx” should be “vortex”.

Author Response

Thanks for your kind reminding.The typo has been corrected in the revised manuscript.